# Applying a cytochrome c oxidase I barcode for *Leishmania* species typing

Carlos Mata-Somarribas[1]*, Grazielle Cardoso das Graças[2], Luiza de Oliveira R. Pereira[3], Mariana Côrtes Boité[3], Lilian Motta Cantanhêde[3], Camila Patrício Braga Filgueira[3], Adrián Fallas[4], Leana Quirós-Rojas[5], Karina A. Morelli[6], Gabriel Eduardo Melim Ferreira[7], Elisa Cupolillo[3]

**1** Centro Nacional de Referencia de Parasitología, Instituto Costarricense de Investigación y Enseñanza en Nutrición y Salud, Cartago, Costa Rica, **2** Hospital Universitário Gaffrée e Guinle–Universidade Federal do Estado do Rio de Janeiro, Rio de Janeiro, RJ, Brasil, **3** Laboratório de Pesquisa em Leishmanioses, Fiocruz—Fundação Oswaldo Cruz, Instituto Oswaldo Cruz, Rio de Janeiro, RJ, Brasil, **4** Hospital Escalante Pradilla, Caja Costarricense de Seguro Social, San José, Costa Rica, **5** Hospital de Guápiles, Caja Costarricense de Seguro Social, Limón, Costa Rica, **6** Departamento de Ecologia, Instituto de Biologia Roberto Alcântara Gomes–Universidade do Estado do Rio de Janeiro, Rio de Janeiro, RJ, Brasil, **7** Laboratório de Epidemiologia Genética, Fundação Oswaldo Cruz Noroeste—Fiocruz Rondônia, Rondônia, RO, Brasil

* cmata@inciensa.sa.cr

## Abstract

Species delimitation has always been a challenge for taxonomists and for *Leishmania* studies there is no exception. Herein we attempt to display the usefulness of the mitochondrial gene Cytochrome Oxidase I–*coI* in classical and barcode-based approaches for *Leishmania* characterization. A total of 228 samples were analyzed, comprising 28 *Leishmania* related taxa, mainly from cultures of the Oswaldo Cruz Foundation's *Leishmania* Collection. Primers were designed for amplification of *coI*; sequences were analyzed by distance-based indicators and both the Neighbor Joining and NeighborNet as species grouping techniques. Automatic Barcode Gap Discovery was applied to define species delimitation while for the character-based analysis a software for Barcoding with Logic formulas was employed. Final sequences of 486 bp with 238 parsimonious sites were aligned and edited. Robust groups were formed for most of the genus species, distinctive nucleotide positions in the barcode sequence were observed for 11 of them. A good agreement between the techniques applied and the original characterization was observed. Few species were not distinguished by *coI*: (i) *L. (V.) peruviana*, *L. (V.) lindenbergi*, and *L. (V.) utingensis*; (ii) *L. (L.) venezuelensis* and (iii) *L. colombiensis* and *L. equatorensis* with identical sequences. Some of these taxa have been, at one time or another, classified as controversial and, for most of them, a higher number of isolates should be studied to properly infer their taxonomic status. *CoI* represents a mitochondrial target that stands out as a taxonomically important asset with multiple advantages over other genes. This paper corresponds to the first report of *coI* analysis in *Leishmania*, a potentially advantageous target for the characterization of this parasite.

**Data Availability Statement:** All DNA sequences are available from the GenBank, NCBI database. Accession numbers are as seen in S1 Table.

**Funding:** CMS received a postgraduate scholarship from IOC as part of his masters' degree. This research was funded by Projeto de Ações Estratégicas e de Fortalecimento - Instituto Oswaldo Cruz PAEF-IOC/Fiocruz, Conselho Nacional de Desenvolvimento Científico e Tecnológico (CNPq; 302622/2017-9) and Fundação de Amparo à Pesquisa do Estado do Rio de Janeiro [FAPERJ; E-26/202.569/2019 (245678)]. PAEF-IOC/FIOCRUZ: https://servicos.fiotec.org.br/PortalAcessoInformacao/DetalheProjeto.aspx?k=1654 CNPq: https://www.gov.br/cnpq/pt-br FAPERJ: https://www.faperj.br/ The authors declare that the funders had no role in study design, data collection and analysis, decision to publish, or preparation of the manuscript.

**Competing interests:** The authors have declared that no competing interests exist.

## Introduction

Defining species delimitation has been a challenge for taxonomists and for *Leishmania* there is no exception. The name *Leishmania* was first proposed for this parasite in 1903 by Ross, referring to the Leishman Donovan bodies previously described in patients' smears suffering from kala-azar [1]. To this day, this protozoan genus has five recognized subgenera that includes around 40 species, more than 20 pathogenic to humans [2]. Many of these species were first described based on ecological, epidemiological and biological characteristics, regarded as extrinsic characters. Former reliance on these extrinsic characters changed after global application of polyphasic taxonomy to diverse groups of organisms, especially bacteria. This method takes into account all available phenotypic and genotypic data and integrates them in a consensus type of classification [3]. Application of these techniques to *Leishmania* was first undertaken with data obtained from isoenzyme electrophoresis, a biochemical technique introduced in the 1970s and used for decades [4]. Multilocus analyses based on enzyme electrophoresis (MLEE) came to be the gold standard for *Leishmania* characterization, and, until now, continues to be the reference for new species identification methods applied for the genus [5, 6].

Despite its contribution, MLEE presents drawbacks, such as the lack of resolution to differentiate gene sequences that share enzymatic phenotypes; or to interpret heterozygous phenotypes, substitutions, post-translational modifications [7, 8], intraspecies variability or convergent evolution [9]. Other important disadvantages include its requirement of bulk cultures, its inability to be applied on clinical samples or uncultured parasites, its labor intensive and time-consuming process, and its limited distribution as a readily available technique, as it can only be performed in a few reference centers [6, 10]. Even in these specialized environments, the request for reference strains to be used in every run as standards, demands additional and continuous flow of reagents, culture mediums, cryopreservation protocols and dedicated technicians, hindering practical application of this technique. Ultimately, the MLEE has never been standardized among the different specialized laboratories as a cross-sectional and homogeneous procedure. Therefore, comparison of results obtained in different laboratories was never possible.

More recently, studies have developed and employed molecular methods for *Leishmania* typing to overcome the above-mentioned MLEE limitations [6]. However, considering the many *Leishmania* species described and/or validated by MLEE, it is essential to properly compare results and thus perform a data-based revision. One common strategy of molecular studies was to use as targets the coding gene sequences of the most relevant enzymes for species identification applied in MLEE. The typing method, based on short sequences from a number of different loci, was named "multilocus sequence typing" or MLST [11]. The clearest advantage of this methodology is the higher resolution, allowing the detection of discrete variation and the identification of relationships between strains. Additionally, the approach offers data portability for sharing results between laboratories in a digital way. For these reasons, MLST has become the method of choice for typing many organisms [12].

Different studies have confirmed the agreement between the MLEE and MLST results [7, 8, 13], also reporting better resolution and congruence [14–16]. However, the lack of databases for *Leishmania*, added to the low number of characterized strains and the lack of consensus on markers have made it difficult to implement MLST as a reference method [17]. For these reasons, the nomenclature "multilocus sequence analysis–MLSA" is most often used in the literature on *Leishmania*, where the word "typing" is replaced by "analysis". The MLSA methodology has proven to be a good tool for the characterization of strains and epidemiological surveillance, as well as for the analysis of population structure and evolutionary studies [18]. The amount of information provided by this technique opened the possibility of making

inferences about the genetic diversity of the evaluated isolates, and the sequences obtained could be used to determine population structures and to point out evidence of recombination events.

Regarding the array of approaches to MLSA, the method has no immediate applicability for routine use but rather for establishing the genus's phylogeny [10]. Its application on clinical samples is restricted due to its limited sensitivity in such samples. Routinely amplifying several genes in parallel and then sequencing them is too costly and time-consuming [6].

The multiplicity of protocols currently available for researchers makes the choice of a particular typing methodology a difficult task. Moreover, the lack of an unequivocal classification scheme is essentially the result of using different markers to study phylogeny and of the lack of a universally applicable species definition [6]. Herein, we aim to contribute to this issue by presenting a simple, cross-laboratorial method to type and define *Leishmania* species through the barcode rational. Barcode is a system designed to ideally provide fast, accurate and automated identifications using species internal markers, providing resolution at the species level. Using gene sequencing, barcoding was intended so that a reliable sequence read can be obtained with a single pass on conventional sequencing platforms [19]. We bring to attention the advantages and usefulness of the mitochondrial gene Cytochrome Oxidase I–*coI*, both, as a single locus marker—referred as barcoding, and as part of a MLSA methodology for *Leishmania* characterization. Recombination disrupts signals of differentiation in the nuclear genome, and mitochondrial DNA is non-recombining, notwithstanding that more studies are needed in this particular subject. Our purpose is to evaluate the potential of *coI* as a barcode for *Leishmania* species identification. For this, we applied a combination of distance-based and character-based methods to assess the performance of *coI* sequences for distinguishing among different *Leishmania* spp.

## Materials and methods

### *Leishmania* strains

A total of 228 strains were analyzed (Table 1), comprising 28 *Leishmania* species or related taxa:

Of these strains, 217 were obtained from cultures of the Oswaldo Cruz Institute's *Leishmania* Collection (*CLIOC*), previously typed by MLEE (S1 Table). The other 12 samples correspond to publicly available sequences of the *coI* target of interest, readily accessible in the GenBank® genomic database of the National Center for Biotechnology Information (NCBI), part of the National Institutes of Health (NIH), located after application of the BLAST tool.

### Cell culture and DNA extraction

Culture strains preserved in liquid nitrogen were thawed and cultivated in Schneider's medium at 25˚ and supplemented with 20% (v/v) heat-inactivated fetal bovine serum to a density of 1 x $10^7$ promastigotes cells / mL (late log phase), as estimated by counting in a Neubauer chamber. The parasites were washed by centrifugation with TE following Internal Standards Protocol (ISP). DNA extraction was performed using the Wizard DNA purification Kit (Promega, Madison, USA) following the manufacturer's instructions.

### PCR protocol

Primers were designed for the conserved region of *coI* of different Trypanosomatidae family members. For this, sequences available at GenBank® for *Trypanosoma brucei* (M14820), *T. cruzi* (FJ203996), *Leishmania* (*L.*) *donovani* (FJ416603), *L.* (*S.*) *tarentolae* (NC000894) and

**Table 1. Number of strains per species and subgenus, corresponding to all available *coI* sequences from *Leishmania* spp strains, analyzed in this study.**

|     | Genus | Subgenus | Specific epithet | Strains (n) |
|-----|-------|----------|------------------|-------------|
| 1   | *Leishmania* | *Leishmania* | *infantum* | 49 |
| 2   |       |          | *amazonensis* | 41 |
| 3   |       |          | *major* | 13 |
| 4   |       |          | *mexicana* | 6 |
| 5   |       |          | *donovani* | 5 |
| 6   |       |          | *venezuelensis* | 4 |
| 7   |       |          | *tropica* | 2 |
| 8   |       |          | *garnhami \| archibaldi \| aethiopica \| turanica* | 1 |
| 9   |       | *Viannia* | *braziliensis* | 34 |
| 10  |       |          | *guyanensis* | 18 |
| 11  |       |          | *panamensis* | 15 |
| 12  |       |          | *shawi* | 6 |
| 13  |       |          | *naiffi \| lainsoni* | 5 |
| 14  |       |          | *peruviana* | 3 |
| 15  |       |          | *lindenbergi \| utingensis* | 1 |
| 16  |       | -        | *hertigi** | 2 |
| 17  |       |          | *deanei** | 2 |
| 18  |       | *Sauroleishmania* | *adleri \| hoogstraali \| tarentolae* | 1 |
| 19  |       | -        | *colombiensis*** | 7 |
| 20  |       |          | *equatorensis*** | 2 |
| 21  | *Endotrypanum* |   | *schaudinni* | 1 |

*Even though Espinosa and collaborators (Espinosa et al., 2018) proposed the subgenus *Porcisia* for these species, other authors considered them belonging to the *Paraleishmania* genus (Kostygov and Yurchenko, 2017). Given these taxonomic incongruences we preferred to just maintain both species as members of the *Leishmania* genus.

**Species not assigned to any subgenus. For more details, see the text.

*Crithidia oncopelti* (X56015) were employed (Fig 1). These primers are not genus-specific, as shown in Fig 1, where *Leishmania* related trypanosomatids were included in the design. The proposed amplicon has a calculated size of 572 bp for all related trypanosomatids analyzed. Primer information is shown on Table 2. Amplification reactions had, for 50 μL total volume, 0.2 μM of each primer, 1.5 mM MgCl$_2$, 0.2 mM deoxyribonucleotide triphosphate (dNTPs), 1U Platinum Taq DNA Polymerase, 1X commercial reaction buffer and approximately 50 ng DNA. Amplification conditions were: 95˚ for 2 minutes, followed by 35 cycles at 95˚ for 30 seconds, 52˚ for 50 seconds and 72˚ for 1 minute, with a final extension at 72˚ for 10 minutes.

## DNA sequencing

The PCR products were purified by "MinElute 96 UF PCR Purification Kit" (Qiagen, Hilden, Germany) following the protocol recommended by the manufacturer. After purification the DNA was quantified in a NanoDrop$^®$. The DNA mass used for sequencing averaged around 7.5 ng, the plate was prepared with 0.2 μM per primer and H$_2$O up to 7.5 μL. Samples were delivered to the DNA Sequencing Platform by Capillary Electrophoresis (Sanger)—RPT01A, part of the Fiocruz's Network of Technological Platforms, for processing in the automatic sequencer "ABI 3730 DNA Analyzer" (Applied Biosystems, CA, USA).

Forward and reverse sequences of *coI* were edited, assembled and merged into consensus sequences using MEGA X (Molecular Evolutionary Genetics Analysis 10.2.2, Pennsylvania State University, PE, USA), and BioEdit (BioEdit Sequence Alignment Editor 7.2.5, Tom Hall,

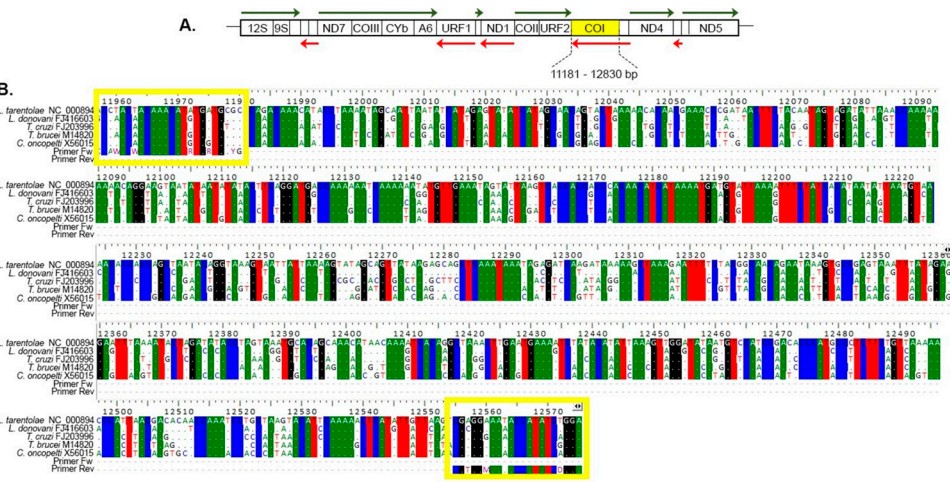

**Fig 1.** Schematic map of the A.) Coding portion of the maxicircle genome of *Leishmania tarentolae* and B.) The multiple sequence alignment of the *coI* gene of related trypanosomatids. A.) Protein coding genes abbreviations are shown inside boxes. Diagram size is proportional to gene size in *Leishmania tarentolae* (Accession Number NC_000894.1). Solely larger genes are shown and its corresponding abbreviations are: 12S and 9S, 12S rRNA and 9S rRNA; ND7, NADH dehydrogenase subunit 7; COIII, cytochrome c oxidase III; CYb, apocytochrome b; A6, ATPase subunit 6; URF1, maxicircle unidentified reading frame 1; ND1, NADH dehydrogenase subunit 1; COII, cytochrome c oxidase II; URF2, maxicircle unidentified reading frame 2; COI, cytochrome c oxidase I; ND4, NADH dehydrogenase subunit 4; ND5, NADH dehydrogenase subunit 5. Green arrows above the boxes represent genes that are transcribed left to right, red arrows below the boxes indicate genes transcribed in the opposite direction. Overall *coI* size is shown under the yellow box, spanning from 11181 to 12830 bp in the *L. tarentolae* published maxicircle genome, the diagram is based on Stuart and Feagin 1992 and Feagin 2000 [20, 21]. B.) The multiple sequence alignment of the *coI* gene between related trypanosomatids and the proposed primers was conducted using the Bioedit package (BioEdit Sequence Alignment Editor 7.2.5, Tom Hall, 1999). The annealing region can be seen marked with a yellow box, for both the forward and reverse primers, with a total length spanning from 11957 to 12575 bp. The final amplification product was calculated around 572 bp. Nucleotides that are highly conserved in these species are shaded with different colors depending on the nucleotide.

1999) packages. Sequences were aligned using MEGA X and BioEdit, applying ClustalW (Conway Institute, University College Dublin, Ireland) parameters.

## Distance-based and clustering analysis

A set of indicators were used to evaluate the performance and discriminating power of the target region (Table 3). The Overall Mean Distance (OMD) is the number of base differences per site obtained from averaging over all sequence pairs [22]. In this case, the OMD was calculated for 228 strains. The Within Group Mean Distance (WGMD) calculates the average number of base differences per site across all sequence pairs within each group, defining as a group a species for which more than one strain has been analyzed. The Between Groups Mean Distance (BGMD) represents the average number of base differences between all pairs of sequences in the proposed groups (S2 Table). The Mean Diversity Within the Population (DWP) is calculated by the number of base differences, per sequence, of the average of the diversity

**Table 2. Amplicon size, primer sequence and location of the *coI* gene target.**

| Gene | Expected product (bp) | Name | Primer Sequence 5'- 3' | Location |
|------|----------------------|------|------------------------|----------|
| ***coI*** | 572 | COI-tripF | CCAWACWACAAACATRTGRTGCYGC | 11181 – 12830bp of *L. (S.) tarentolae* maxicircle |
| | | COI-tripBR | TCCHGATATGGTATTKCCACG | |

**Table 3. Comparison of distance-based indicators data between *Leishmania* species to evaluate the performance and discriminatory power of the target region.**

| Groups | Distance and diversity analysis | | | | | | | |
|---|---|---|---|---|---|---|---|---|
| | WGMD | SE | OMD | SE | DWP | SE | DWS | SE |
| *L. (V.) braziliensis* | 0.005 | 0.002 | 0.126 | 0.009 | 0.123 | 0.009 | 0.007 | 0.001 |
| *L. (V.) peruviana* | 0.001 | 0.001 | | | | | | |
| *L. (V.) panamensis* | 0.001 | 0.001 | | | | | | |
| *L. (V.) guyanensis* | 0.001 | 0.001 | | | | | | |
| *L. (V.) shawi* | 0.000 | 0.000 | | | | | | |
| *L. (V.) naiffi* | 0.005 | 0.002 | | | | | | |
| *L. (V.) lainsoni* | 0.004 | 0.002 | | | | | | |
| *L. (L.) mexicana* | 0.000 | 0.000 | | | | | | |
| *L. (L.) venezuelensis* | 0.045 | 0.007 | | | | | | |
| *L. (L.) amazonensis* | 0.049 | 0.004 | | | | | | |
| *L. (L.) major* | 0.006 | 0.002 | | | | | | |
| *L. (L.) donovani* | 0.000 | 0.000 | | | | | | |
| *L. (L.) infantum* | 0.000 | 0.000 | | | | | | |
| *L. (L.) tropica* | 0.000 | 0.000 | | | | | | |
| *L. colombiensis* | 0.000 | 0.000 | | | | | | |
| *L. equatorensis* | 0.000 | 0.000 | | | | | | |
| *L. hertigi* | 0.000 | 0.000 | | | | | | |

WGMD: Within group mean distance, SE: Standard Error, OMD: Overall mean distance, DWP: Mean diversity within the population, and DWS: Mean diversity within subpopulations.

calculations, for the entire population and for any given subpopulations (DWS) [23]. These diversity calculations were generated automatically by MEGA X, applying equation number 12.73. The standard error of these calculations was obtained by the "bootstrap" method (1000 replicates). The codon positions included were 1st + 2nd + 3rd + Noncoding, and all ambiguous positions were removed for each pair of sequences (pairwise deletion).

Species groupings, or "clusters", were obtained by building trees using the Neighbor Joining (NJ) method [24], which is not concerned with grouping the most closely related Operational Taxonomic Units, but rather with minimizing the size of all internal nodes and thus the size of the entire tree. The *p*-distance (proportion of nucleotide sites at which two sequences are different) was computed to infer the differences in nucleotide composition between sequences. Node support was evaluated with 1000 bootstrap replicates. Bootstrap equal or above 90% were considered for groups definition.

Strain clusters were also obtained by the NeighborNet method, which proposes networks based on a distance matrix given by the Uncorrected-P or Hamming distance, in which, for any two sequences of equal length, the Hamming distance is the number of positions at which the corresponding symbols are different [25]. For this goal the Splitstree program (SplitsTree4 4.17.1, Eberhard Karls University of Tübingen, Tübingen, Germany) was used.

## Automatic Barcode Gap Discovery (ABGD)

To evaluate the barcode species delimitation, the ABGD method (Automatic Barcode Gap Discovery, web version 08/26/21) was applied to the DNA sequences obtained from the range of *Leishmania* species. This method statistically infers the "Barcode Gap" of the sequence data and partitions the species accordingly. When differences between sequence pairs are compared in a dataset, it is possible to observe a "gap" that separates intraspecific and interspecific

diversity. *CoI* sequence data was processed in ABGD using simple distance, the value of intra-specific divergence was set between 0.001 and 0.1, with 10 recursive steps and a gap width set of 1.5.

### Character-based analysis

To evaluate the barcode sequence proposed herein, the BLOG software (Barcoding with LOGic formulas v2, Istituto di Analisi dei Sistemi e Informatica, Rome, Italy) was applied. This method performs a character-based analysis that identifies, for each species in a reference library, the distinctive nucleotide positions of the DNA barcode sequence, and assigns to each species classification formulas that can tightly characterize a species [26].

## Results

Herein, we gathered and analyzed nucleotide sequences covering the mitochondrial *coI* gene, representing 228 *Leishmania* related strains. These strains had been previously identified at the species level by MLEE and/or other molecular method (S1 Table). The strains included, accounted for 1 *Leishmania* related parasite (Genus *Endotrypanum*—*"nomen dubium")*, 3 *Leishmania* subgenera (*Viannia*, *Leishmania*, *Sauroleishmania)* and 27 *Leishmania* species.

### Characteristics of *coI* sequences

PCR products of around 570bp were analyzed after *coI* amplification. Final sequences of 486 nucleotides were obtained after alignment and editing for all *Leishmania* species, showing 213 conserved sites, 273 variable, 35 singleton and 238 parsimonious sites. *CoI* sequences from the sampled strains showed an A + T bias (average = 68,5) relative to the C + G content (average = 31,5). The average content of each nucleotide was as follows: T = 22.9, C = 19.5, A = 45.6, and G = 12.0.

### Distance-based analysis

The overall mean *p*-distance between sequences, OMD, was 0.126, ranging from 0.0 to 0.049 within groups, WGMD (Table 3). The lowest average between groups, BGMD, was 0.000, between *L. colombiensis* and *L. equatorensis*, and, in non-*Viannia* species, was 0.015 between *L. (L.) infantum* and *L. (L.) donovani*, while the highest was 0.200 between *L. colombiensis/L. equatorensis* and *L. (L.) mexicana*. Considering only *L. (Viannia)* species, the lowest *p*-distance average was 0.004, between *L. (V.) braziliensis* and *L. (V.) peruviana*, and 0.018, between *L. (V.) panamensis* and *L. (V.) shawi*, while the highest were observed between *L. (V.) lainsoni* and *L. (V.) shawi* (0.054), and *L. (V.) lainsoni* and *L. (V.) guyanensis* (0.052) (S2 Table).

 After performing the NJ analysis and running 1000 replications with p-distance calculation, the generated tree clustered a large group with a high bootstrap value (100) which corresponds to the subgenus *L. (Viannia)* (Fig 2). In this cluster, we observed the formation of sub-groups that correlate, for the most part, with the analyzed species, with the exception of *L. (V.) peruviana*, *L. (V.) utigensis* and *L. (V.) lindenbergi*, which grouped together with sequences of *L. (V.) braziliensis*. Sequences corresponding to subgenus *L. (Sauroleishmania)* form a cluster with high support (100). For the genus *Endotrypanum* and those *Leishmania* species not assigned to any subgenus, a particular cluster was observed, with two branches, one including *L. deanei* and *L. hertigi* and the other including *L. colombiensis* and *L. equatorensis*.

 The NJ tree also indicated that sequences from the Old-World species, classified in the subgenus *L. (Leishmania)*, formed a robust group with subgroups clearly related to the analyzed species (Fig 3). It is important to highlight that *coI* allows the identification of species of the *L.*

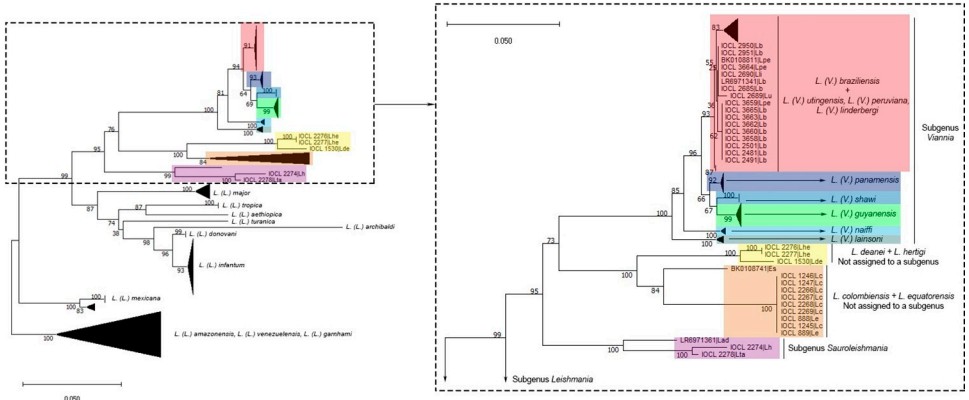

**Fig 2. Species grouping, obtained from the Neighbor Joining (NJ) method, corresponding to the *L. (Viannia)* subgenus, for the *coI* gene analysis.** On the left, the original NJ tree can be observed with all the analyzed species. On the right, the cutout shows *L. (Viannia)* species clustering. Color shading was applied to simplify tracking from the original tree.

*(L.) donovani* complex, although only one sequence of *L. (L.) archibaldi* was analyzed and few of *L. (L.) donovani*. Sequences referring to the *L. (L.) mexicana* complex formed two groups. One group with high support is composed of *L. (L.) amazonensis* from La Paz, Bolivia, two *L. (L.) venezuelensis* from Lara, Venezuela, and all of *L. (L.) mexicana*. The other group was formed by the two other *L. (L.) venezuelensis* from Lara, Venezuela, *L. (L.) garnhami* and the remaining *L. (L.) amazonensis* from South America, mainly from Brazil.

The NeighborNet analysis (Fig 4) provided clear identification of species clusters agreeing with the original species characterization. As in the NJ tree, NeighborNet *splits* show a clear definition of species clusters for the *L. (Viannia)* and *L. (Sauroleishmania)* subgenus and for *L. deanei* and *L. hertigi*. For the *L. (Leishmania)* subgenus, defined clusters for *L. (L.) major*, *L. (L.) donovani*, *L. (L.) infantum*, *L. (L.) tropica*, *L. (L.) aethiopica*, *L. (L.) turanica* and *L. (L.) archibaldi* can be observed, but interpretation remains a challenge for the *L. (L.) mexicana* complex, particularly when it comes to *L. (L.) amazonensis*.

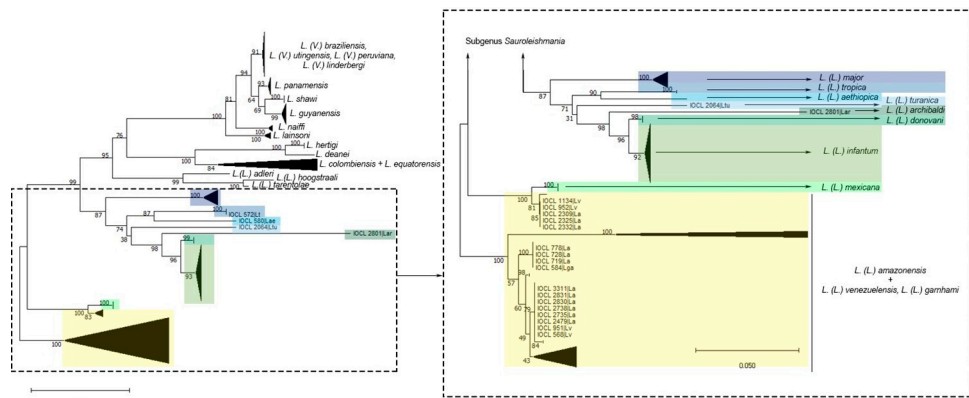

**Fig 3. Species grouping, obtained from the Neighbor Joining (NJ) method, corresponding to the *L. (Leishmania)* subgenus, for the *coI* gene analysis.** On the left, the original NJ tree can be observed with all the analyzed species. On the right, the cutout shows *L. (Leishmania)* species clustering. Color shading was applied to simplify tracking from the original tree.

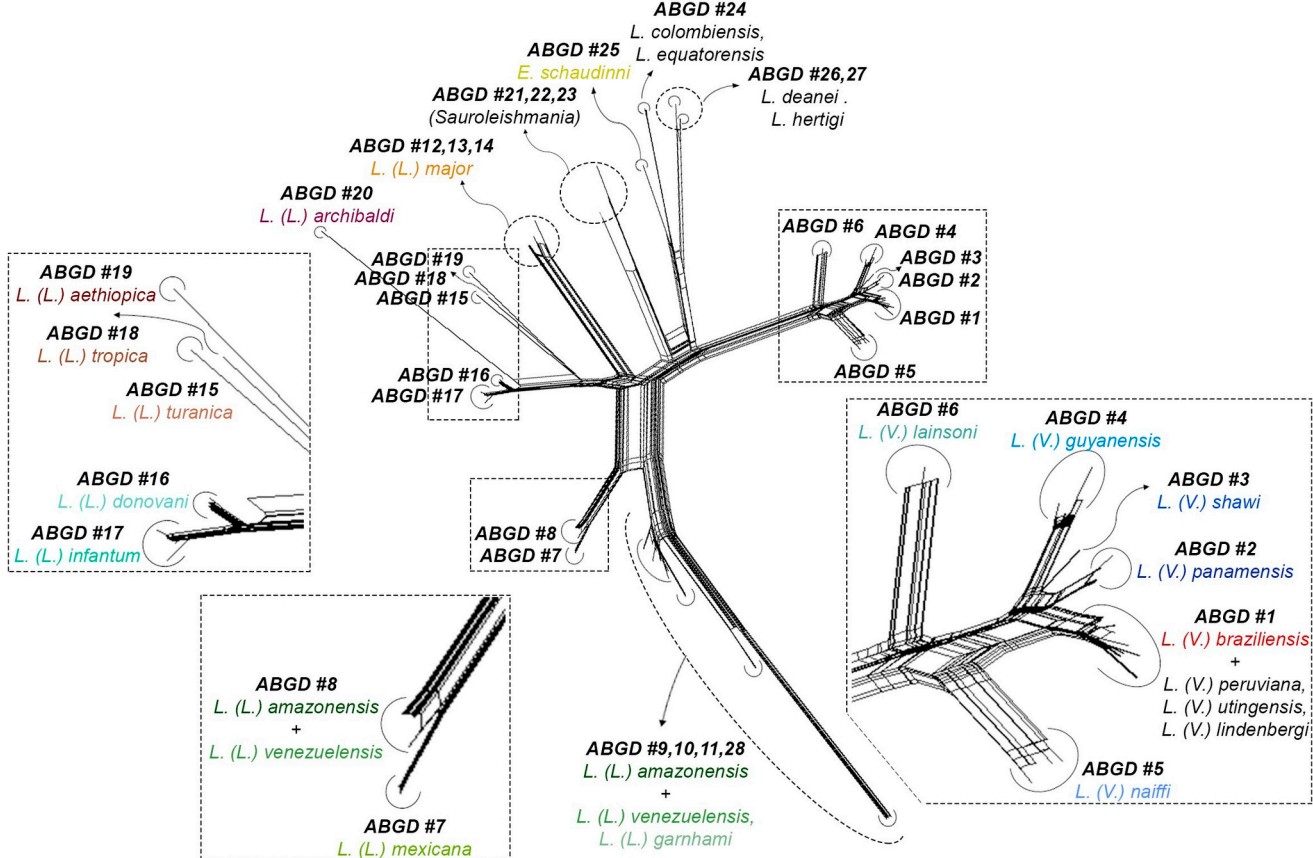

**Fig 4. Species clustering obtained by the NeighborNet analysis with its matching ABGD grouping, for all *coI* sequences analyzed.** NeighborNet *splits* can be shown graphically separating groups of species. Over the NeighborNet *splits* the distinct ABGD groups were labelled, showing agreement between different analysis. Segments of the NeighborNet *splits* were cutout and zoomed in, for visualization ease. Color lettering was applied to simplify species identification.

## Automatic Barcode Gap Discovery (ABGD)

Applying ABGD, nine values of "prior intraspecific divergence" (P) were evaluated to identify potential barcode gaps. Considering P values less than 0.005, since values above this threshold tend to underestimate the number of species, 28 partitions were observed for this data set (Table 4). The ABGD algorithm agreed with the previous characterization of each sample, separating species into 28 groups. The subgenus *L. (Viannia)* was partitioned from the rest of samples and *L. (V.) panamensis*, *L. (V.) lainsoni*, *L. (V.) naiffi* and *L. (V.) shawi* were each identified as distinct groups. However, *L. (V.) braziliensis* continued to be grouped with *L. (V.) linderbergi*, *L. (V.) peruviana* and *L. (V.) utingensis*. As previously indicated for other analysis the barcode gap failed to identify some species from the *L. (L.) mexicana* complex, although it allowed the separation of *L. (L.) mexicana* from *L. (L.) venezuelensis* and *L. (L.) amazonensis*. A high intraspecific variability was observed again in *L. (L.) amazonensis*, which was classified into five distinct groups. For the species not classified in any subgenera, distinct ABGD groups were observed: *L. colombiensis* and *L. equatorensis* were separated from other species as one group, while *L. hertigi* and *L. deanei* were partitioned individually.

**Table 4. Species groups formed by partitions made with Automatic Barcode Gap Discovery (ABGD) for the *coI* gene.**

| Group | Samples |
|---|---|
| ABGD #1 | IOCL_1009\|Lb IOCL_1734\|Lb IOCL_1906\|Lb IOCL_1907\|Lb IOCL_2950\|Lb IOCL_2951\|Lb IOCL_2481\|Lb IOCL_2491\|Lb IOCL_1943\|Lb IOCL_2073\|Lb IOCL_2122\|Lb IOCL_2125\|Lb IOCL_2160\|Lb IOCL_2889\|Lb IOCL_2918\|Lb IOCL_2927\|Lb IOCL_3072\|Lb IOCL_3090\|Lb IOCL_2463\|Lb IOCL_2468\|Lb IOCL_2475\|Lb IOCL_2476\|Lb IOCL_2515\|Lb IOCL_2872\|Lb IOCL_2501\|Lb IOCL_3658\|Lb IOCL_3659\|Lpe IOCL_3660\|Lb IOCL_3662\|Lb IOCL_3663\|Lb IOCL_3665\|Lb IOCL_2685\|Lb IOCL_2690\|Lli IOCL_3664\|Lpe LR6971341\|Lb BK0108811\|Lpe IOCL_2689\|Lu IOCL_3239\|Lb IOCL_3451\|Lb |
| ABGD #2 | IOCL_1252\|Lpa MK5705101\|Lpa BK0108751\|Lpa IOCL_240\|Lpa IOCL_3284\|Lpa IOCL_3285\|Lpa IOCL_3286\|Lpa IOCL_3287\|Lpa IOCL_243\|Lpa IOCL_3288\|Lpa IOCL_3293\|Lpa IOCL_3294\|Lpa IOCL_3297\|Lpa IOCL_3301\|Lpa IOCL_3302\|Lpa |
| ABGD #3 | IOCL_1066\|Ls IOCL_1068\|Ls IOCL_1545\|Ls IOCL_3481\|Ls IOCL_1067\|Ls BK0108831\|Ls |
| ABGD #4 | IOCL_2334\|Lg IOCL_2335\|Lg IOCL_2336\|Lg IOCL_2337\|Lg IOCL_2341\|Lg IOCL_2356\|Lg IOCL_2372\|Lg IOCL_2389\|Lg IOCL_2396\|Lg IOCL_2398\|Lg IOCL_2405\|Lg IOCL_2410\|Lg IOCL_2692\|Lg IOCL_2963\|Lg IOCL_2338\|Lg IOCL_2371\|Lg IOCL_3013\|Lg IOCL_565\|Lg |
| ABGD #5 | IOCL_1365\|Ln IOCL_3310\|Ln IOCL_3007\|Ln IOCL_993\|Ln IOCL_854\|Ln |
| ABGD #6 | IOCL_1023\|Lla IOCL_1266\|Lla BK0108791\|Lla IOCL_3315\|Lla IOCL_2497\|Lla |
| ABGD #7 | IOCL_1015\|Lm IOCL_1020\|Lm IOCL_117\|Lm IOCL_1227\|Lm IOCL_561\|Lm IOCL_577\|Lm |
| ABGD #8 | IOCL_1134\|Lv IOCL_2309\|La IOCL_2325\|La IOCL_2332\|La IOCL_952\|Lv IOCL_2307\|La |
| ABGD #9 | IOCL_1043\|La IOCL_1060\|La IOCL_1071\|La IOCL_1730\|La IOCL_2024\|La IOCL_930\|La IOCL_575\|La IOCL_2459\|La IOCL_2571\|La IOCL_2479\|La IOCL_2735\|La IOCL_2738\|La IOCL_2830\|La IOCL_2831\|La IOCL_3281\|La IOCL_3311\|La IOCL_568\|Lv IOCL_951\|Lv IOCL_59\|La IOCL_71\|La IOCL_584\|Lga IOCL_719\|La IOCL_728\|La IOCL_729\|La IOCL_778\|La IOCL_2730\|La IOCL_2737\|La IOCL_2740\|La IOCL_704\|La IOCL_876\|La IOCL_897\|La IOCL_589\|La |
| ABGD #10 | IOCL_2586\|La IOCL_2674\|La IOCL_2679\|La |
| ABGD #11 | IOCL_324\|La IOCL_536\|La |
| ABGD #12 | IOCL_1044\|Lma IOCL_1231\|Lma IOCL_133\|Lma IOCL_183\|Lma IOCL_1840\|Lma IOCL_186\|Lma IOCL_51\|Lma IOCL_574\|Lma IOCL_581\|Lma IOCL_867\|Lma IOCL_895\|Lma |
| ABGD #13 | IOCL_2798\|Lma |
| ABGD #14 | IOCL_2728\|Lma |
| ABGD #15 | IOCL_2064\|Ltu |
| ABGD #16 | IOCL_2272\|Ld IOCL_2800\|Ld IOCL_563\|Ld CP0226521\|Ld FJ4166031\|Ld |
| ABGD #17 | IOCL_2301\|Li IOCL_2906\|Li IOCL_2990\|Li IOCL_3034\|Li IOCL_3049\|Li IOCL_3053\|Li IOCL_3249\|Li IOCL_579\|Li IOCL_656\|Li IOCL_3068\|Li IOCL_3132\|Li IOCL_3203\|Li IOCL_3204\|Li IOCL_3206\|Li IOCL_3208\|Li IOCL_3219\|Li IOCL_3226\|Li IOCL_3233\|Li IOCL_3253\|Li IOCL_3254\|Li IOCL_3256\|Li IOCL_3257\|Li IOCL_3328\|Li IOCL_3330\|Li IOCL_3331\|Li IOCL_3333\|Li IOCL_3335\|Li IOCL_3336\|Li IOCL_3337\|Li IOCL_3338\|Li IOCL_3339\|Li IOCL_3340\|Li IOCL_3341\|Li IOCL_3342\|Li IOCL_3343\|Li IOCL_3344\|Li IOCL_3346\|Li IOCL_3348\|Li IOCL_3368\|Li IOCL_3369\|Li IOCL_3376\|Li IOCL_3377\|Li IOCL_3378\|Li IOCL_3379\|Li IOCL_3381\|Li IOCL_3384\|Li IOCL_3385\|Li IOCL_3388\|Li MT7622871\|Li |
| ABGD #18 | IOCL_571\|Lt IOCL_572\|Lt |
| ABGD #19 | IOCL_580\|Lae |
| ABGD #20 | IOCL_2801\|Lar |
| ABGD #21 | IOCL_2274\|Lh |

*(Continued)*

**Table 4.** (Continued)

| Group | Samples |
|---|---|
| *ABGD #22* | IOCL_2278\|Lta |
| *ABGD #23* | LR6971361\|Lad |
| *ABGD #24* | IOCL_1245\|Lc IOCL_1246\|Lc IOCL_1247\|Lc IOCL_2266\|Lc IOCL_2267\|Lc IOCL_2268\|Lc IOCL_2269\|Lc IOCL_888\|Le IOCL_889\|Le |
| *ABGD #25* | BK0108741\|Es |
| *ABGD #26* | IOCL_1530\|Lde |
| *ABGD #27* | IOCL_2276\|Lhe IOCL_2277\|Lhe |
| *ABGD #28* | IOCL_185\|La IOCL_262\|La IOCL_621\|La |

Lb: L. (V.) braziliensis, Lpe: L. (V.) peruviana, Lli: L. (V.) lindenbergi, Lu: L. (V.) utingensis, Lpa: L. (V.) panamensis, Ls: L. (V.) shawi, Lg: L. (V.) guyanensis, Ln: L. (V.) naiffi, Lla: L. (V.) lainsoni, Lm: L. (L.) mexicana, Lv: L. (L.) venezuelensis, La: L. (L.) amazonensis, Lga: L. (L.) garnhami, Lma: L. (L.) major, Ltu: L. (L.) turanica, Ld: L. (L.) donovani, Li: L. (L.) infantum, Lt: L. (L.) tropica, Lae: L. (L.) aethiopica, Lar: L. (L.) archibaldi, Lh: L. (S.) hoogstraali, Lta: L. (S.) tarentolae, Lad: L. (S.) adleri, Lc: L. colombiensis, Le: L. equatorensis, Es: E. schaudinni, Lde: L. deanei, Lhe: L. hertigi.

## Character-based analysis

The BLOG software was used to obtain logic formulas, also known as character-based patterns, for each species. These formulas revealed unique and unequivocal nucleotide positions in the DNA barcode sequence for 11 of the 17 taxa that had more than one sequence per species. The other 11 species, from the 28 taxa analyzed, only reported one sequence per species, and so, invalidated the BLOG software logic formulas. As shown in Table 5, the 11 species with a distinctive nucleotide positioning, of the 17 analyzed, had a 0.00 false positive rate for those diagnostic positions. Nevertheless, while analyzing these results it came to attention that two taxa showed a lower Laplacian Score. This score, which allows to rank features based on its locality preserving property [27], was notably low for *L. (L.) tropica* and *L. hertigi*, at just 0.069. After analyzing all related sequences, it was found that the distinctive nucleotide positions for *L. (L.) tropica* were not unique to that species. These positions were also present in *L. (L.) aethiopica* and *L. (L.) archibaldi*. The same occurred with *L. hertigi*, which shared its character pattern with *L. deanei*.

For another three species, from the group of 17 that had more than one sequence analyzed, *L. (V.) braziliensis*, *L. (L.) amazonensis* and *L. colombiensis*, the BLOG software was able to identify diagnostic characters but with a false positive rate > 0.00. The remaining three species of the selected group, *L. (V.) peruviana*, *L. (L.) venezuelensis* and *L. equatorensis*, did not show any distinctive nucleotide positioning.

## Discussion

*CoI* nucleotide sequences of a representative set of *Leishmania* species were subjected to classical and barcode-based analysis. Results suggest the target represents a good species marker, considering the high agreement between the ABGD groups, distance-based analysis, monophyletic clades, species clusters and the original characterization by MLEE or molecular identification.

**Table 5. Specific nucleotide positions that identify each *Leishmania* species, confirming a *barcode* for *coI* sequences analysis.**

|  | Species | Positions | Nucleotides | False positive rate | Laplace Score |
|---|---|---|---|---|---|
| **1** | *L. (V.) panamensis* | 15 AND 423 | T AND C | 0.000 | 0.325 |
| **2** | *L. (V.) shawi* | 15 AND 423 | C AND C | 0.000 | 0.156 |
| **3** | *L. (V.) guyanensis* | 96 AND 423 | C AND T | 0.000 | 0.357 |
| **4** | *L. (V.) naiffi* | 15 AND 483 | C AND T | 0.000 | 0.156 |
| **5** | *L. (V.) lainsoni* | 423 AND 483 | C AND T | 0.000 | 0.156 |
| **6** | *L. (L.) mexicana* | 56 AND 315 | T AND T | 0.000 | 0.156 |
| **7** | *L. (L.) major* | 15 AND 423 AND 483 | C AND A AND G | 0.000 | 0.289 |
| **8** | *L. (L.) donovani* | 87 AND 315 | T AND C | 0.000 | 0.156 |
| **9** | *L. (L.) infantum* | 87 AND 96 | C AND G | 0.000 | 0.597 |
| **10** | *L. (L.) tropica* | 15 | G | 0.000 | 0.069 |
| **11** | *L. hertigi* | 108 AND 483 | G AND C | 0.000 | 0.069 |
| **12** | *L. (V.) braziliensis* | 96 AND 423 | A AND T | 0.014 | 0.491 |
| **13** | *L. (L.) amazonensis* | 486 OR 108 AND 483 | C OR G AND A | 0.007 | 0.517 |
| **14** | *L. colombiensis* | 15 AND 96 | A AND A | 0.006 | 0.176 |

A: adenine, T: thymine, C: cytosine, G: guanine.

However, a few limitations were found, such as within the *L. (V.) braziliensis* complex. *L. (V.) peruviana*, *L. (V.) lindenbergi* and *L. (V.) utingensis* clustered with *L. (V.) braziliensis* strains. These strains were grouped in the same phylogenetic clade, as seen in the NJ analysis (Fig 2). No nucleotide differences were detected between *L. (V.) peruviana* IOCL-3659 and eight *L. (V.) braziliensis* strains. Interestingly, these strains were all but one from Peru and from Acre, a nearby region of the Brazilian Amazon. However, the other two *L. (V.) peruviana* showed a few differences when compared to *L. (V.) peruviana* IOCL-3659. These two strains displayed no nucleotide differences when compared to *L. (V.) lindenbergi* and four *L. (V.) braziliensis* strains not related to the Amazon region. The *L. (V.) utingensis* sequence was slightly different from these other sequences with an equal or inferior intraspecific variability than that within *L. (V.) braziliensis* strains.

Regarding *L. (V.) peruviana*, its status as a species has been questioned many times before [28]. Nevertheless, different analysis based on nuclear targets have seemed to show that *L. (V.) braziliensis* and *L. (V.) peruviana* are two closely-related species but distinct, monophyletic lines [29]. The close relationship between these two species involves very similar or indistinguishable genetic targets. Analyses of the mitochondrial Cytochrome b gene (*cytB*) conducted by independent laboratories were unable to differentiate between both species [30–32], corroborating the *coI* findings. Few studies have been conducted on *L. (V.) lindenbergi* and *L. (V.) utingensis* [33, 34], but they form separate and independent branches and share almost no alleles with other species, thus corroborating MLEE data [18]. More isolates from both species should be studied to properly infer their taxonomic status. Within the *L. (L.) mexicana* complex, a clear barcode gap was found for *L. (L.) mexicana* but, curiously, *L. (L.) amazonensis* strains were split in five clusters, with three of them showing identical nucleotide sequences between different species: i) two *L. (L.) venezuelensis* and seven *L. (L.) amazonensis*; ii) two other *L. (L.) venezuelensis* and three *L. (L.) amazonensis*); iii) one *L. (L.) garnhami* and four *L. (L.) amazonensis*. *L. (L.) amazonensis* showed the largest WGMD (Table 3), which was also demonstrated in the NJ and NeighborNet clustering, and in the barcode partition which resulted in five different groups of strains.

In this species complex there is still a controversial debate on the status of several taxa. *L. (L.) venezuelensis* has been described as a valid species on the basis of distinguishable MLEE

patterns and specific monoclonal antibodies reactions [35]. Moreover, the occurrence of a heterogeneous population has been reported for this species [36]. However, molecular data is limited, and some authors strongly suggested *L. (L.) venezuelensis* is actually a variant of *L. (L.) mexicana* [37]. The current *coI* findings indeed reveal a great heterogenicity among these samples, but they appear more related to the *L. (L.) amazonensis* strains, which also show high variability. Therefore, *L. (L.) venezuelensis* taxonomic status remains inconclusive.

*L. (L.) pifanoi* and *L. (L.) garnhami* status has been debated practically since its determination as species [38], but more recently it has been widely accepted that they correspond to synonyms of *L. (L.) mexicana* and *L. (L.) amazonensis*, respectively [39, 40].

An argument for inconclusiveness regarding its status can also be made for *L. (L.) waltoni*, a more recent taxon described from the Dominican Republic [41], and for *L. (L.) ellisi*, a species isolated from an autochthonous case in Arizona, United States [42]. In the former case, an isolated group of strains from diffuse cutaneous leishmaniases lesions [43] were determined to be more closely related to *L. (L.) mexicana*, although showing biologically similarities with *L. (L.) venezuelensis*. In the latter, a single strain from a papular cutaneous lesion was isolated and determined to be closely related to *L. (L.) mexicana* and *L. (L.) amazonensis* [44]. These examples exhibit the need for a better understanding of this species complex and the designation of better markers that allow a clear definition of species. More robust molecular information seems to be needed, as it is questionable that all the species of this complex deserved specific status, and that no subspecies can be defined [45].

For the controversial taxa *L. colombiensis/L. equatorensis*, both were undistinguishable from each other in their *coI* sequence (S2 Table). Taxonomy regarding species close to the *Endotrypanum* genus has been debatable for decades. *L. colombiensis* seems to be more closely related to that genus than its resembling counterpart, *L. equatorensis*, and more importantly, it has been frequently associated with human disease [46]. *L. equatorensis* has just recently been isolated from human lesions in Colombia [47] but overall, there are few reports of natural infection by these species in other Latin American countries, either in humans or other animal reservoirs [48]. *L. equatorensis* and *L. colombiensis* comprise a distinct group within the Leishmaniinae subfamily based on minicircle analysis [49], monoclonal antibodies, enzyme electrophoretic analysis, restriction fragment analysis, schizodeme analysis and more recent molecular assays [50–54]. The similarity between these two peculiar species has been demonstrated before, and were confirmed by the present *coI* analysis, in which no differences between the samples could be detected.

Based on these results and previous arguments [51], we maintain *Endotrypanum* as a "*nomem dubium*", rejecting the proposal to classify *L. colombiensis* and *L. equatorensis* in that genus [46]. Furthermore, the results obtained cannot be used to assign *L. hertigi* and *L. deanei* to a separate genus [46], as this would imply a revision of the *Leishmania* classification, elevating other subgenera to new genera. Without a doubt, this is a picture of how complex *Leishmania* taxonomy really is and reinforces the need for an urgent revision, employing various markers and shared strains to be analyzed by different research groups.

Selecting the most appropriate species typing technology in a given context can be a challenging task [6]. Several approaches have been used to identify *Leishmania* species, including the analysis of different regions of the genome, and numerous targets have been sequenced. Identification of *Leishmania* species is an essential step for epidemiological studies that must address the etiological agent, but it is also pivotal for genetic diversity research in clinical scenarios, given its recognized clinical importance. For instance, in the Americas, several species of *Leishmania* influence clinical manifestations, severity of the disease, accuracy of diagnosis and response to treatment [55].

The structure of *Leishmania*'s mitochondrial genome (maxicircle and minicircles) has been characterized, shedding light on particularly useful phylogenetic targets as for example *cytB*. This gene has been described as a high sensitivity, high specificity uniparentally informative mitochondrial target [47, 56–59]. *CytB* gene has been included in different MLST sets currently used for phylogenetic studies in *Leishmania* [60].

As with *cytB*, *coI* represents another mitochondrial taxonomically valuable target. Since the proposal of "DNA Barcoding", it has been widely used in barcode studies of various organisms [19]. A number of characteristics support that mitochondrial genes are good candidates to be used as species markers: rare insertions and deletions [61], offering stability–a desirable trait for species markers. Also, their high copy number that could increase sensitivity in PCR based approaches and their different mutation rates between species [62], that admits for molecular evolutionary based species typing. This marker apparently do not undergo recombination, although in *Leishmania* this still deserves more studies [63]; it has an assumed haploid inheritance [64] and has low ancestral polymorphism with high conservation between strains [65]. Despite the advantages and the fact that it has already been used for characterization of related pathogens as *Trypanosoma cruzi* [66], this target has not yet been fully applied to the identification of the *Leishmania* protozoan. *CoI* represents a good mitochondrial sequencing target to be added to the arsenal of targets already available for the identification of *Leishmania* species. *CoI* shows high discriminatory power and robustness in all analyses, relevant polymorphisms and significant informative sites between evaluated species. Herein we showed *coI* barcoding being successfully applied to species typing.

The present study corresponds to the first report of *coI* analysis in *Leishmania*. The few available sequences from public databases were included herein, in addition to those originally obtained by us, comprising a total of 228 sequences. Despite the number of DNA sequences, the number of strains *per* species was limited, partially restricting the analysis.

An additional limitation is the quality of sequences deposited in public databases. These are not always originated from taxonomic studies with rigorous postanalytical control compromising the overall quality of the information associated with *Leishmania* sequences. It is not uncommon to find data with sequencing blunders, wrong nomenclature, lack of biological or molecular information, misidentified species and identification errors in deposited material. For this reason also, using a standardized and curated database of *Leishmania* strains should be prioritize before use of other repositories. The scientific community involved in *Leishmania* taxonomy should work towards a standardized and curated database of strains, with the objective of simplifying phylogeny and taxonomy studies, consensus should be the final goal.

The application of mitochondrial targets for *Leishmania* typing shows cases of divergence when pertaining to New World *Leishmania* species [30]. This disagreement indicates a mismatch between kinetoplast and nuclear genes, known as mito-nuclear discordance, already reported in isolates from Latin America, specifically observed for the *L. (Viannia)* subgenus [31]. Further study is needed to disclose the mechanisms involved [67], which possibly include genetic exchange and the formation of hybrids in the nuclear genome, and thus, a mito-nuclear discordance. The frequency and mechanisms are still unclear, but it may be present in unexpectedly high rate (~10%) [68]. These findings invalidate any proposal that applies just one mitochondrial target for typing purposes, or application of just one target for typing, either nuclear or mitochondrial.

Finally, we encourage the scientific community to apply *coI* directly to clinical samples and address species characterization with this target. Although the protocol described herein could be of limited use because of the final product size, additional primers could be proposed based on the now available *coI* sequences. Evaluation of *coI* barcoding for *Leishmania* species

identification showed great potential. Nevertheless, considering the existing limitations of typing *Leishmania* species with just one single locus marker, *coI* should be considered as a first line target for MLSA or MLST sets, incorporating both nuclear and mitochondrial genes.

## Supporting information

**S1 Table. DNA sequences obtained from GenBank®, of the National Center for Biotechnology Information (NCBI), and from the Oswaldo Cruz Foundation's Leishmania Collection database, corresponding to all available *coI* sequences from Leishmania spp. strains, for the evaluation of the *coI* target.** IOC-L is the prefix use by the Oswaldo Cruz Institute's Leishmania Collection (CLIOC) as part of the deposit code applied to Leishmania strains. (DOCX)

**S2 Table. Between Groups Mean Distance (BGMD), a distance-based indicator, showing the average number of base differences of all sequence pairs, between the proposed species groups.** (DOCX)

## Acknowledgments

To local laboratories of the Caja Costarricense de Seguro Social and their work regarding surveillance of parasitological diseases, to Ms. Wendy Valverde, Ms. Mariel Chaves and Ms. Stephanie Quirós for their technical support, to Dr. Marlon Gregori and Dr. Franklyn Samudio for their timely answers and recommendations, to the Programa de Pós-graduação em Biologia Celular e Molecular and to the Laboratório de Pesquisa em Leishmanioses do Instituto Oswaldo Cruz (IOC)–Fundação Oswaldo Cruz (Fiocruz).

## Author Contributions

**Conceptualization:** Carlos Mata-Somarribas, Mariana Côrtes Boité, Elisa Cupolillo.

**Data curation:** Carlos Mata-Somarribas, Grazielle Cardoso das Graças, Luiza de Oliveira R. Pereira, Camila Patrício Braga Filgueira, Adrián Fallas, Leana Quirós-Rojas, Karina A. Morelli, Gabriel Eduardo Melim Ferreira.

**Formal analysis:** Carlos Mata-Somarribas, Grazielle Cardoso das Graças, Luiza de Oliveira R. Pereira, Mariana Côrtes Boité, Lilian Motta Cantanhêde, Camila Patrício Braga Filgueira, Karina A. Morelli, Gabriel Eduardo Melim Ferreira, Elisa Cupolillo.

**Funding acquisition:** Elisa Cupolillo.

**Investigation:** Mariana Côrtes Boité, Elisa Cupolillo.

**Methodology:** Carlos Mata-Somarribas, Mariana Côrtes Boité, Elisa Cupolillo.

**Project administration:** Elisa Cupolillo.

**Resources:** Mariana Côrtes Boité, Elisa Cupolillo.

**Supervision:** Luiza de Oliveira R. Pereira, Mariana Côrtes Boité, Elisa Cupolillo.

**Writing – original draft:** Carlos Mata-Somarribas.

**Writing – review & editing:** Carlos Mata-Somarribas, Luiza de Oliveira R. Pereira, Mariana Côrtes Boité, Lilian Motta Cantanhêde, Camila Patrício Braga Filgueira, Adrián Fallas, Leana Quirós-Rojas, Karina A. Morelli, Gabriel Eduardo Melim Ferreira, Elisa Cupolillo.

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
