## [Decision Letter · Decision Letter 0]

3 Sep 2024

PONE-D-24-33743Applying a cytochrome c oxidase I barcode for *Leishmania* species typingPLOS ONE

Dear Dr. Mata-Somarribas,

Thank you for submitting your manuscript to PLOS ONE. After careful consideration, we feel that it has merit but does not fully meet PLOS ONE’s publication criteria as it currently stands. Therefore, we invite you to submit a revised version of the manuscript that addresses the points raised during the review process.

Be sure to:Indicate which changes you require for acceptance versus which changes you recommendAddress any conflicts between the reviews so that it's clear which advice the authors should followProvide specific feedback from your evaluation of the manuscript==============================

We look forward to receiving your revised manuscript.

Kind regards,

Felipe Dutra-Rêgo, PhD

Academic Editor

PLOS ONE

Reviewers' comments:

Reviewer's Responses to Questions

**Comments to the Author**

1. Is the manuscript technically sound, and do the data support the conclusions?

Reviewer #1: Yes

Reviewer #2: Yes

2. Has the statistical analysis been performed appropriately and rigorously? 

Reviewer #1: Yes

Reviewer #2: Yes

3. Have the authors made all data underlying the findings in their manuscript fully available?

Reviewer #1: Yes

Reviewer #2: Yes

4. Is the manuscript presented in an intelligible fashion and written in standard English?

Reviewer #1: Yes

Reviewer #2: Yes

5. Review Comments to the Author

Reviewer #1: In this manuscript, the authors describe the use of the mitochondrial gene Cytochrome Oxidase I for Leishmania typing. The manuscript is clear and well-written. I have some suggestions that I consider that may improve the quality of this study.

Main points:

1) It is not clear in the manuscript whether the primers designed for amplification of conserved region of coI are able to amplify coI gene of Trypanosomatidae family members, as Trypanosoma cruzi, T. brucei. Could you clarify this point in the manuscript. Are these primers specific for Leishmania genus?

2) In Table 2, it is indicated that the amplicon size is 486 bp, while in Results, the amplicon has around 600bp. Please, include the expected size of this amplicon in Table 2. Are there differences in size of this amplicon among Leishmania species? If not, what are the variations in size? Could you include this information in the manuscript?

3) I would also suggest to include a figure in the manuscript containing the ORF of coI gene and the location of these primers.

Reviewer #2: In the article “Applying a cytochrome c oxidase I barcode for Leishmania species typing” the authors study the COI gene as a molecular marker for phylogenetic analysis and characterization of Leishmania species in order to build a genetically supported taxonomy for the entire genus and obtain a marker for Leishmania species identification. Therefore, it is necessary to agree on a practical taxonomic classification scheme based on reliable and consensual concepts. A good definition of Leishmania species is crucial for the correct diagnosis and prognosis of the disease, as well as for making decisions about treatment and control measures. The article contribute to the taxonomic discussion of Leishmania genus and provide and other potential gene for Leishmania species identification. The useful of this marker for the detection and identification directly for clinical samples will be study in a future. The article is very well writing and the results obtained are very well discussed.

6. PLOS authors have the option to publish the peer review history of their article (what does this mean?). If published, this will include your full peer review and any attached files.

Reviewer #1: No

Reviewer #2: **Yes: **Jorge Fraga Nodarse

---

## [Author Response · Author response to Decision Letter 0]

25 Sep 2024

We wish to submit the corresponding response to the questions and comments stated by the reviewers to our research article entitled “Applying a cytochrome oxidase I barcode for Leishmania species typing” for consideration by Plos One: 

Reviewer #1: In this manuscript, the authors describe the use of the mitochondrial gene Cytochrome Oxidase I for Leishmania typing. The manuscript is clear and well-written. I have some suggestions that I consider that may improve the quality of this study.

Main points:

1) It is not clear in the manuscript whether the primers designed for amplification of conserved region of coI are able to amplify coI gene of Trypanosomatidae family members, as Trypanosoma cruzi, T. brucei. Could you clarify this point in the manuscript. Are these primers specific for Leishmania genus?

Response: The primers were designed to amplify the family Trypanosomatidae, they are not specific to Leishmania. We avoided designing a genus-specific primer in order to generate lines of research, in the near future, involving different members of the family. For example, one paper of some colleagues of ours has already been published, applying the same primers to T. cruzi with a few modifications. We will clarify this situation in the manuscript, so there’s no confusion. We hope to keep working with this Leishmania research line in order to improve and optimize the proposed PCR protocol.

2) In Table 2, it is indicated that the amplicon size is 486 bp, while in Results, the amplicon has around 600bp. Please, include the expected size of this amplicon in Table 2. Are there differences in size of this amplicon among Leishmania species? If not, what are the variations in size? Could you include this information in the manuscript?

Response: We took the opportunity given by Reviewer #1’ last recommendation and propose a schematic map of the gene and its corresponding multiple alignment. In this figure we thoroughly explain the characteristics of the amplicon for which we have a calculated size of 572 bp for all related trypanosomatids analyzed. After processing, eliminating the ends and clearing low quality regions, Leishmania species consistently show a 486 bp amplicon, suitable for genetic sequencing. 

3) I would also suggest to include a figure in the manuscript containing the ORF of coI gene and the location of these primers.

Response: Following this recommendation, we included a new figure in the manuscript with all corresponding information regarding coI gene.

Reviewer #2: In the article “Applying a cytochrome c oxidase I barcode for Leishmania species typing” the authors study the COI gene as a molecular marker for phylogenetic analysis and characterization of Leishmania species in order to build a genetically supported taxonomy for the entire genus and obtain a marker for Leishmania species identification. Therefore, it is necessary to agree on a practical taxonomic classification scheme based on reliable and consensual concepts. A good definition of Leishmania species is crucial for the correct diagnosis and prognosis of the disease, as well as for making decisions about treatment and control measures. The article contribute to the taxonomic discussion of Leishmania genus and provide and other potential gene for Leishmania species identification. The useful of this marker for the detection and identification directly for clinical samples will be study in a future. The article is very well writing and the results obtained are very well discussed.

Response: We are thankful to Reviewer #2 for his kind words regarding this article, an effort that took years in the making, and that we finally get to published.

---

## [Editor Report · Decision Letter 1]

30 Sep 2024

Applying a cytochrome c oxidase I barcode for *Leishmania* species typing

PONE-D-24-33743R1

Dear Dr. Mata-Somarribas,

We’re pleased to inform you that your manuscript has been judged scientifically suitable for publication and will be formally accepted for publication once it meets all outstanding technical requirements.

Kind regards,

Felipe Dutra-Rêgo, PhD

Academic Editor

PLOS ONE
---

## [Editor Report · Acceptance letter]

4 Oct 2024

PONE-D-24-33743R1 

PLOS ONE

Dear Dr. Mata-Somarribas, 

I'm pleased to inform you that your manuscript has been deemed suitable for publication in PLOS ONE. Congratulations! Your manuscript is now being handed over to our production team.

Kind regards, 

on behalf of

Dr. Felipe Dutra-Rêgo 

Academic Editor

PLOS ONE